# Performance study of oil-injection hermetic CO$_2$ scroll compressor for automotive air conditioning system

**Jingying Hao**[1]*, **Zhizhong Wang**[2], **Gaoxuan Bu**[3], **Bo Zhang**[4], **Pengcheng Shu**[5]

**1** School of Energy and Environmental Engineering, Hebei University of Engineering, Handan, P. R. China, **2** National Engineering Research Center of Fluid Machinery and Compressors, Xi'an Jiaotong University, Xi'an, P. R. China, **3** National Engineering Research Center of Fluid Machinery and Compressors, Xi'an Jiaotong University, Xi'an, P. R. China, **4** College of Energy Engineering, Xi'an University of Science and Technology, Xi'an, P. R. China, **5** National Engineering Research Center of Fluid Machinery and Compressors, Xi'an Jiaotong University, Xi'an, P. R. China

* bigai06228@163.com

**Data Availability Statement:** All relevant data are within the manuscript and its Supporting Information files.

**Funding:** The work described in the paper was funded by the Program for Changjiang Scholars

## Abstract

An oil-injection CO$_2$ scroll compressor prototype and relative simulation model are developed for automotive Air Conditioning. The effects of oil to refrigerant mass ratio on the compressor performance are studied theoretically and experimentally. The results show that oil-injection is an effective way to improve the volumetric and indicated efficiencies, and reduce discharge temperature. The optimal oil-injection quantity varies at different conditions, and it decreases from 7.9% to 6.3% at fixed discharge pressure 10MPa when the suction pressure increases from 4.5MPa to 5MPa. Similar linear relationships are found and the different slopes of the line are exhibited at fixed suction and discharge pressure separately. Numerical results show that oil-injection has little influence on pressure in working chamber during suction and discharge process, but it can slow up the rising of pressure in compression process and reduce the salient of pressure at the end of compression process.

## 1. Introduction

In order to replace the environmentally harmful refrigerant CFCs and HCFCs, one of the most important challenges is to seek a suitable refrigerant in refrigeration and Air Conditioning industries [1,2]. Natural refrigerant carbon dioxide exhibits unique characteristics, such as: excellent heat transfer properties, larger volumetric refrigerating capacity, non-flammable and non-toxic features, so it is considered as the most ideal alternative refrigerants by Lorentzen G [3]. Since the transcritical CO$_2$ cycle, which can overcome the low critical temperature of CO$_2$, was presented by Lorentzen G in the late of last century, CO$_2$ has been gained increasing attentions [4].

Many researches focus not only on the performance of transcritical CO$_2$ system [4–6], but also on the 4 main components, especially on the key part—CO$_2$ compressor. Among the different types of compressors, CO$_2$ scroll compressor has been continually studied in the last

and Innovative Research Team in University (IRT0746), Key Technology R&D Program (2009BAA17B00) and the Program of Guangdong province (2007B090400079) The funders had no role in study design, data collection and analysis, decision to publish, or preparation of the manuscript.

Abbreviations: $A$, Flow area (m$^2$); $C_d$, Flow coefficient; $C_v$, Specific heat at constant specific volume (J·kg$^{-1}$·K$^{-1}$); $F$, Force (N); $h$, Specific enthalpy of the refrigerant in the control volume (J·kg$^{-1}$); $h_1$, Enthalpy of refrigerant at suction state based on the measured value of temperature and pressure (J·kg$^{-1}$); $h_{2s}$, Discharge enthalpy when the compression process is assumed to be isentropic one (J·kg$^{-1}$); $m$, Mass of refrigerant (kg); $\dot{m}_s$, Designed mass flow rate (kg·s$^{-1}$); $\dot{m}_{sim}$, Simulated mass flow rate (kg·s$^{-1}$); $\dot{m}_{exp}$, Experimentally measured mass flow rate (kg·s$^{-1}$); $n$, Compressor speed (rev·min$^{-1}$); $N$, Friction loss (W); $P$, Pressure of refrigerant (Pa); $P_E$, Electric power input into compressor motor (W); $Pr$, Prandtl number; $Q$, Heat flow rate into the control volume (W); $R$, Gas constant/moment of force (m); $Re$, Reynolds number; $R_s$, Radius of the crank pin (m); $R_{mj}$, Radius of the main bearing (m); $r$, The orbiting radius (m); $T$, Temperature of refrigerant (K); $v$, Specific volume of refrigerant (m$^3$·kg$^{-1}$); $v_{th}$, Theoretical calculated displacement volume (m$^3$); $v_{act}$, Experimental measured displacement volume (m$^3$); $v_s$, Designed displacement volume(m$^3$); $V$, Volume of the control volume (m$^3$); Greek symbols $\delta_a$, Axial clearance; $\delta_r$, Radial clearance; $\varepsilon$, Pressure ratio; $\varepsilon_{cr}$, Critical pressure ratio; $\eta_{i-sim}$, Simulated indicated efficiency; $\eta_{i-exp}$, Experimentally measured indicated efficiency; $\eta_{v-sim}$, Simulated volumetric efficiency; $\eta_{v-exp}$, Experimentally measured volumetric efficiency; $\theta$, Crank angle (rad); $\kappa$, Specific heat ratio; $\tau$, Time (s); $\rho_s$, Density of the suction gas (kg·m$^{-3}$); $\omega$, Angular speed of compressor crank (rad·s$^{-1}$) or Solubility of $CO_2$ in lubrication oil; Subscripts g, Gas; l, Liquid; in, Flow in; out, Flow out; up, inlet of the nozzle; down, outlet of the nozzle; d, discharge; s, suction; oil, Lubrication oil; lr, Liquid refrigerant; leak, leakage through the axial and radial clearance.

decade due to its merits of higher durability, reliability, efficiency, lower noise and smaller vibration.

Fagerli BE [7] demonstrated that the gas leakage was a very knotty problem in his theoretical analysis of the $CO_2$ automotive Air Conditioning scroll compressor in 1998. Compared with the conventional one, if the equal efficiency could be achieved, the theoretical axial and radial clearance gap must be 5μm. So a self-adjusting back-pressure mechanism was proposed to assure this small gap. Comparative studies between $CO_2$ and R134a scroll compressor were conducted by Hirao T [8]. The simulation results showed that smaller pressure loss, larger leakage loss and lower mechanical efficiency occurred in $CO_2$ scroll compressor. After some improving measures were adopted, such as orbiting scroll pushed to fixed scroll, rolling bearing replaced by thrusting bearing, the adiabatic efficiency of the $CO_2$ scroll compressor prototype was 76% at 2400 RPM. Ishii N [9] calculated the efficiency of a compact $CO_2$ scroll compressor and compared it with same cooling capacity of R410A compressor. It was found that leakage loss was also the main factor of lower resultant efficiency for $CO_2$ scroll compressor. The volumetric efficiency for 3μm-axial and 6μm-radial clearances was lower by 5%, therefore resulting in the resultant efficiency of 4% lower, although the mechanical and compression efficiency were higher by about 1%. Yoshida H [10] designed several new scroll profiles to reduce leakage loss, when the temperature distribution of fixed and orbiting scroll was measured and the uneven thermal expansion data was achieved. Then 1% of the resultant efficiency improvement could be obtained compared with the conventional scroll compressor. Hasegawa H [11] fabricated a small capacity hermetic $CO_2$ scroll compressor prototype and then it was analyzed experimentally and theoretically. The conclusions presented that the volumetric efficiency of the prototype was lower than that of a R410A scroll compressor due to the large pressure difference between and suction and discharge. The major loss came from the thrust bearing. Hao JY [12] fabricated a $CO_2$ hermetic scroll compressor prototype with a high-pressure vessel. Experimental results showed that the maximum volumetric efficiency and indicated efficiency are 84% and 80%, respectively. Hiwata A [13] paid special attention to control the problem of orbiting scroll overturning. It was found that a new groove made on the thrust bearing of the fixed scroll could prevent the overturning efficiently, even in the low-compression ratio operation conditions. Yano K [14] developed a new thrust bearing configuration with flexible structure when the very high thrust load of $CO_2$ scroll compressor was considered. In the experiments, the new thrust structure worked effectively and reliably, it can improve the scroll compressor efficiency by 2%. Hiwata A [15] developed a $CO_2$ scroll compressor with the fixed-radius crank mechanism and calculated the contact force between the wraps on the basis of measuring the main bearing oil-film pressure. The fluctuation of the contact force at one revolution was calculated and the effects of rotational speed and wrap clearance ratio on the contact force were analyzed. Zheng SY [16–18] calculated the working process of $CO_2$ scroll compressor by considering the radial and tangential refrigerant leakage, the CFD simulated results indicated that the Pre-compression at the end of suction process and the Over-compression at the end of compression process were obviously observed. The tangential leakage model simulated results indicated that the seal-grooves at compression chamber could increase the volumetric efficiency and isentropic efficiency by 1.63% and 1.32% separately, and the seal-grooves at suction chamber could improve the volumetric efficiency by 0.99%. the radial leakage model simulated results showed that the volumetric and isentropic efficiency could raise up 2.1% and 1.0% when the groove width was 0.5 mm and the groove depth was 100 μm.

The above reviews present that the leakage through axial and radial clearance gap has a significant influence on performance of $CO_2$ scroll compressor because of high pressure difference. But Literatures mainly focus on decrease the clearance gaps to reduce leakage loss, but

the clearance gaps less than 10μm are difficult to be guaranteed in operation. Few studies concentrated on the effect of oil -injection into $CO_2$ scroll compressor, although oil-injection is an effective measure of reducing leakage, increase the volumetric efficiency and reduce the discharge temperature. An oil flooded R410A scroll compressor was tested at the hot gas bypass stand and its semi-empirical thermodynamic model were established, experimental results showed that the volumetric efficiency was increased from 93% of no oil injection to about 98% at oil mass fraction 0.5, when the test condition was set as evaporation temperature -10˚C and condensation temperature 43.3˚C [19–21]. Wang C [22] developed a helium oil-flooded scroll compressor, the experimental results indicate that the discharge temperature can be remarkably lowered from 455˚C to approximately 64˚C with the oil injection into working chamber. The volumetric efficiency was sharply increased from 66% of no oil injection to about 93.6% at the oil flow rate 6 L/min. Wang HL [23] established a finite element model of the $CO_2$ orbiting scroll at design condition of suction pressure 4MPa and discharge pressure 10MPa, simulated results indicated that the maximum deformation displacement of the gear head, radical and tangential was 7.33 μm, 4.42 μm and 5.89μm respectively. Only Hiwata A and Sawai K [24,25] carried out the work of oil injecting to $CO_2$ scroll compressor. They designed an intermittent oil supply mechanism to control the oil flow rate to the compression chambers and developed the prototype of oil-injection $CO_2$ scroll compressor with high pressure shell. By testing the prototype, the optimum oil-injection rates at different working conditions were obtained. In additional, a leakage index of the compressor by the dimensional analysis was derived, which was proportional to optimal oil flow rate. But the oil supply system was very complex.

In this paper, in order to solve the working process internal leakage of $CO_2$ scroll compressor at the bigger differential pressure about 5 and 6 MPa, an oil supply control system is designed, which has effective distribution function in lubricating friction pairs and oil-injection to working chamber. A prototype of oil-injection $CO_2$ scroll compressor is fabricated for automotive Air Conditioning. Experimental investigation on the prototype has been carried out at different conditions in order to test the effect of oil-injection on the prototype performance, so the change of volumetric efficiency, indicated efficiency and discharge temperature with respect to mass ratio of oil to refrigerant were conducted, the values of optimal oil-injection quantity at different working conditions were obtained. Moreover, a numerical simulation model involving gas leakage, and heat transfer is established and verified experimentally. Simulation results of prototype working process with different mass ratio of oil to refrigerant were presented.

## 2. Basic structure of the prototype

The cross section of the oil-injection $CO_2$ hermetic scroll compressor prototype is shown in Fig 1. It is designed to be low pressure shell to ensure safety in operation and reduce the total machine weight. A fixed scroll, an orbiting scroll, a three-phase AC motor, an oil pump are located from up to down in the prototype shell. A crank shaft close couples with the stator through its hole and transmits the rotation of the motor to the orbiting scroll. A main bearing and a sub bearing support the crank shaft tightened in the frame. A ball bearing anti-rotation mechanism was used to prevent the self-rotation of orbiting scroll. The suction is designed at the lower position of the vessel and discharge port is located at the central location of the upper end plate. The $CO_2$ refrigerant flows into the suction port, enters into the working chamber through the motor's gaps and is discharged out of the compressor directly.

An oil supply system contains a lubrication part and an oil-injection part. The cycloidal gear oil pump, which is connected with the crank shaft using a key, is immersed in the oil pool at the bottom of the vessel. As shown in Figs 1 and 2, the oil is pumped out of the compressor

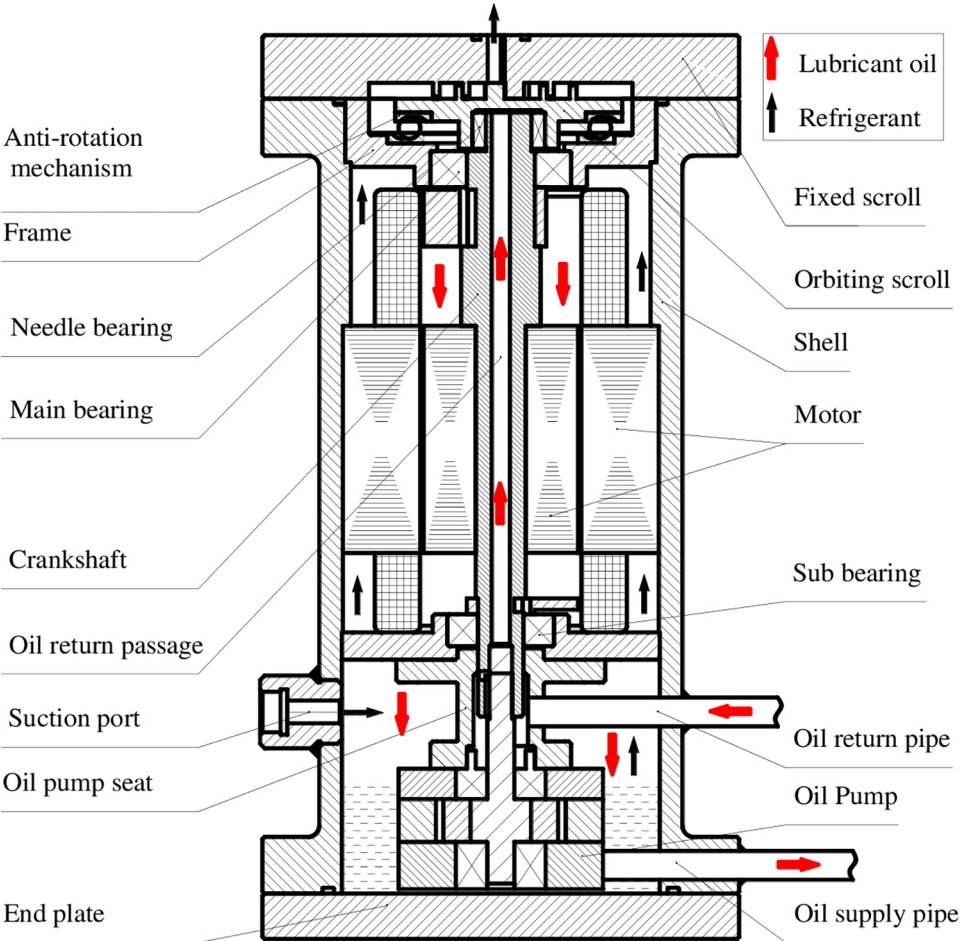

**Fig 1. Schematic diagram of chamber oil-injection $CO_2$ hermetic scroll compressor.**

into the pipe line PP2 and separated into two fluids at point P2, one return back into the lubrication part via the pipe line P2P5, lubricates the main bearing and the sub bearing passing through the hole inside the crank shaft, and move back to the bottom of the vessel via the gaps inside the motor. The other fluid flows along the pipe line P2P8 and is injected into the two suction chambers by the pipe lines P9P11 and P10P12. The pipe lines P9P11 and P10P12 have the same dimensions to assure equal injection quantity. In order to adjust and monitor the oil quality injected into the compressor, two needle valves (2, 4), two sintered glass sight (3, 5), one ball valve (6), three pressure gauges (7.1–7.4) are arranged in the stainless steel oil pipeline. The needle valves 2 and 4 are used to be co-adjusted to change the distribution of the two separated oil. Then different oil-injection quantities into the compressor can be realized. The total cycle number of the needle valve 2 and 4 is 6.5 from closed to full opening position.

The photo of the prototype compressor and the oil-injection control mechanism are presented in Fig 3. In the picture, large yellow arrows represent the oil flow and the refrigerant flow is expressed by small yellow arrows. As shown in Fig 4, the two oil-injection ports are arranged symmetrically in the fixed scroll at rotation angle 40˚, its diameter was designed as 0.6mm. The two oil-injection connectors $P_{11}$ and $P_{12}$ are connected with the fixed scroll by sealing thread.

The structural parameters of the prototype are listed in Table 1. The perfect mesh profile (PMP) has been adopted to modify the tip profile of the $CO_2$ scroll compressor, which can

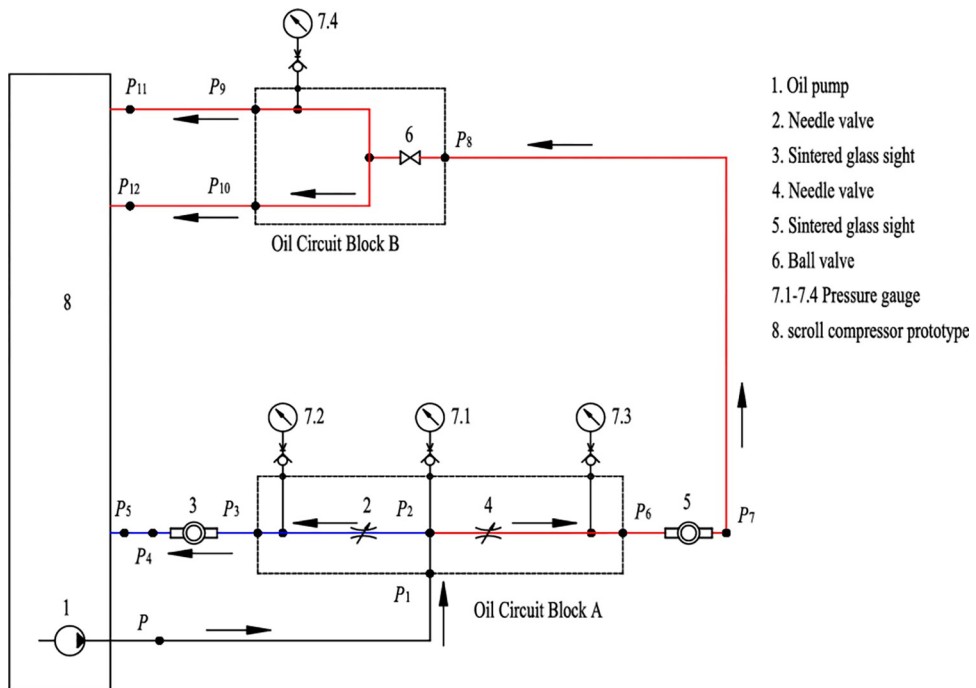

**Fig 2. The chamber oil-injection control system principle diagram.**

reduce the leakage loss at the end of the compression process and increase the strength of the scroll wrap. The designed value of axial and radial gap was 15 μm by considering the assembly precision and the effect of oil-injection to working chamber.

## 3. Theoretical model

A numerical simulation model is established to predict the working process of the $CO_2$ scroll compressor prototype, and the model is consists of a geometrical model, a leakage model and a heat transfer model. The geometrical model describes the change of crescent working chamber volume with respect to the rotation angle. The leakage model simulates the leakage loss of the mixture of refrigerant and lubricant oil though the axial and redial clearance gaps. The heat transfer model calculates the heat transfer between the mixture and the fixed scroll plate, orbiting scroll plate, previous and next working chambers. The working process governing equations of the $CO_2$ scroll compressor is derived from the mass, energy conservation equations and the real gas state equations.

### 3.1 Governing equations

For the given control volume, the homogeneous flow model is adopted. The differential equation of gas temperature ($T_g$) with respect to the rotation angle is given as below [26].

$$\frac{\partial T_g}{\partial \theta} = \frac{1}{m_g C_{vg}} \left\{ -T \left(\frac{\partial P}{\partial T}\right)_V \left[\frac{\partial V}{\partial \theta} - \frac{v_g}{\omega}\left(\dot{m}_{gin} - \dot{m}_{gout}\right)\right] - \sum \frac{\dot{m}_{gin}}{\omega}\left(h_g - h_{gin}\right) + \frac{Q_{gin}}{\omega} \right\} \quad (1)$$

The gas mass balance is described by the following equation [26].

$$\frac{\partial m_g}{\partial \theta} = \sum \frac{\dot{m}_{gin}}{\omega} - \sum \frac{\dot{m}_{gout}}{\omega} \quad (2)$$

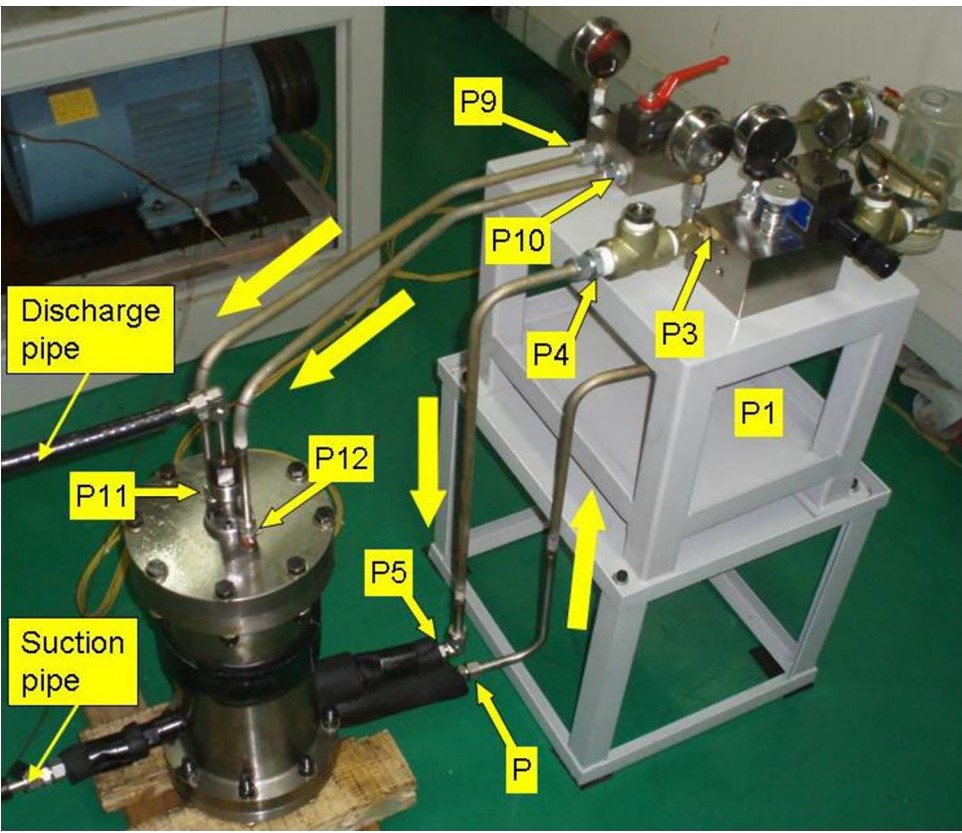

**Fig 3. Photo of the $CO_2$ scroll compressor prototype and its oil-injection control system.**

The differential equation of liquid temperature ($T_l$) with respect to the rotation angle is given as follows [26].

$$\frac{\partial T_l}{\partial \theta} = \frac{1}{m_l C_l} \left\{ \frac{dQ_{lin}}{d\theta} + \sum \frac{dm_{lin}}{d\theta} h_{lin} - \sum \frac{dm_{lout}}{d\theta} h_l - C_l T_l \frac{dm_l}{d\theta} \right\} \tag{3}$$

The liquid mass balance equation [26] is shown as follows.

$$\frac{\partial m_l}{\partial \theta} = \sum \frac{\dot{m}_{lin}}{\omega} - \sum \frac{\dot{m}_{lout}}{\omega} \tag{4}$$

In the above equations, $dQ_{gin} = (1\text{-}z)dQ_{in}$, $dQ_{lin} = zdQ_{in}$ and $z = (m_{oil}+m_{lr})/(m_r+m_{oil})$.

The PR state equation [27] and the Vander Waals-Berthelot mixture rules were used to calculate the solubility of $CO_2$ in lubricant. The state of $CO_2$ and lubricant oil mixture is calculated by the following modification correlation equations [26].

$$T = \frac{m_g C_{vg} T_g + m_l C_{vl} T_l}{m_g C_{vg} + m_l C_{vl}} \tag{5}$$

$$P = P_g = P_l = P(T, v_g) \tag{6}$$

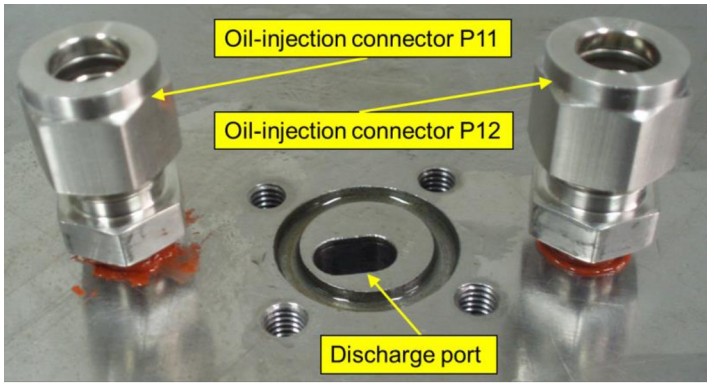

(a)

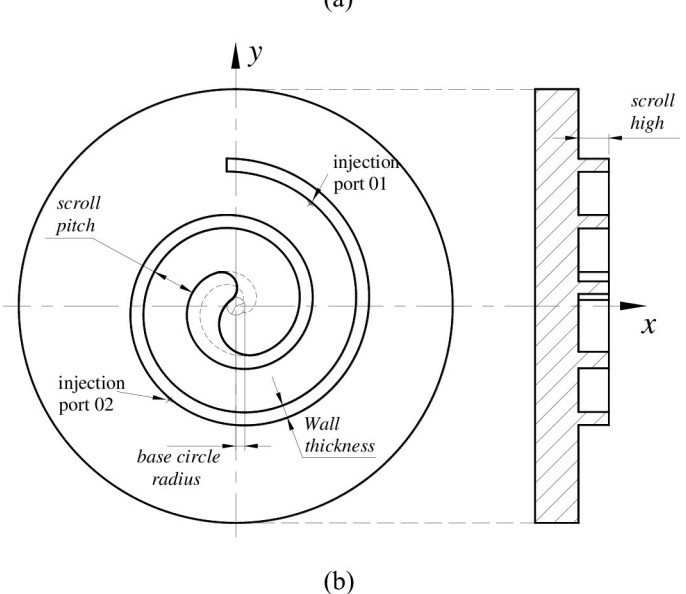

(b)

**Fig 4. Photo of the oil-injection ports on the fixed scroll.**

**Table 1. Parameters of the prototype.**

| Items | Value |
|---|---|
| Cooling capacity / kW | 5.0 |
| Displacement volume /cc | 7.0 |
| Involute base circle radius / mm | 2.07 |
| Scroll height / mm | 7.0 |
| Scroll thickness /mm | 3.0 |
| Axial clearance($\delta_a$) /μm | 15.0 |
| Radial clearance($\delta_r$) /μm | 15.0 |
| Dimension(D×H) /mm×mm | Φ160×398 |

## 3.2 Leakage model

The leakages through the axial gaps and the radial gap are computed by nozzle model on the hypothesis of one-dimensional compressible flow and adiabatic process [28]

$$\frac{dm}{d\tau} = C_d A P_{up} \sqrt{\frac{2k}{R(k-1)T_{up}}(\varepsilon^{\frac{2}{k}} - \varepsilon^{\frac{k+1}{k}})} \tag{7}$$

$$\varepsilon_{cr} = \left(\frac{2}{k+1}\right)^{\frac{k}{k-1}} \tag{8}$$

When $\frac{P_{down}}{P_{up}} \geq \varepsilon_{cr}, \varepsilon = \frac{P_{down}}{P_{up}}; \frac{P_{down}}{P_{up}} < \varepsilon_{cr}, \varepsilon = \varepsilon_{cr}.$

## 3.3 Heat transfer model

In order to simulate the compression process accurately, a heat transfer model is established. The temperature $T_w$, which is defined as a function of the involute angle $\varphi$, is assumed as the linear distribution along the scroll wraps from the involute angle $\varphi_{os}$ at the center of the scroll to the involute angle $\varphi_e$ at the outer edge of the scroll [29]

$$T_w = \bar{T} + \frac{T_d - T_s}{\varphi_e - \varphi_{os}}(\varphi - \frac{\varphi_e + \varphi_{os}}{2}) \tag{9}$$

Where $\bar{T}$ is the average temperature of scroll wrap at the involute angle $\frac{\varphi_e + \varphi_{os}}{2}$.

The convective heat transfer coefficient is calculated by spiral tube modified Dittus-Boelter equation [29].

$$h_c = 0.023(\frac{\lambda}{D_h})\mathrm{Re}^{0.8}\mathrm{Pr}^{0.4}(1.0 + 1.77\frac{D_h}{r_{aver}}) \tag{10}$$

Where $D_h$ denotes the equivalent hydraulic diameter, and $r_{aver}$ stands for the average radius of the curved chamber calculated by the following equation [29].

$$r_{aver} = a\left[\frac{(\varphi_k - \pi/2) + (\varphi_{k-1} - \pi/2)}{2}\right] \tag{11}$$

Where $\varphi_k$ and $\varphi_{k-1}$ are the involute angles at *kth* node and (k-1)th node of conjugacy, respectively.

The total heat transfer rate can be calculated by the following equation.

$$Q = \sum h_c(\theta)A_h(\theta)\Delta T_w(\theta) \tag{12}$$

## 3.4 Performance parameters

The simulated volumetric efficiency and the experimentally measured volumetric efficiency are defined as follows [30].

$$\eta_{v-sim} = \frac{\dot{m}_{sim}}{\dot{m}_s} \tag{13}$$

$$\eta_{v-exp} = \frac{\dot{m}_{exp}}{\dot{m}_s} \tag{14}$$

The simulated mass flow rate is determined in the form of the following equation.

$$\dot{m}_{sim} = \frac{\omega}{2\pi}\int \dot{m}_\theta d\theta \tag{15}$$

Designed mass flow rate is defined as following [31].

$$\dot{m}_s = \frac{v_s \cdot \rho_s \cdot n}{60} \tag{16}$$

The simulated and experimentally measured indicated efficiencies are defined by the following equations [30].

$$\eta_{i-sim} = \frac{\dot{m}_{sim} \cdot (h_{2s} - h_1)}{P_{i-sim}} \tag{17}$$

$$\eta_{i-exp} = \frac{\dot{m}_{exp} \cdot (h_{2s} - h_1)}{P_{i-exp}} \tag{18}$$

The simulated indicated power is given by

$$P_{i-sim} = \frac{\omega}{2\pi}\int VdP \tag{19}$$

The indicated power measured by experiment is expressed as the product of the power input to the motor, the motor efficiency and the mechanical efficiency, the motor efficiency calibration was given by motor manufacturer, the mechanical friction loss and flow loss at different working conditions was calculated by using the empirical formula for friction coefficient [32]. In the empirical formula, the force $F$ was the force value of main bearing, sub bearing and crank pin respectively, which was obtained by the dynamic Characteristic analysis of Scroll Compressor [33], the value of friction coefficient was given by the volume 2 of mechanical design manual.

$$N_{friction} = \frac{\omega}{2\pi}\int \mu \cdot (F \cdot R)d\theta \tag{20}$$

$$P_{i-exp} = P_E \eta_{motor} \eta_m \tag{21}$$

The calculated discharge temperature is defined as the integration average value of the temperature during the whole discharge process [29].

$$T_d = \frac{1}{2\pi}\sum_{i=1}^{Max} T_i(\theta)\Delta\theta_i \tag{22}$$

## 3.5 Numerical solution

Fig 5 shows the flow chart of the oil-injection $CO_2$ scroll compressor model. At first, geometric size and operation conditions are taken as input parameters, and the modified angle $\beta$ and the flow area of discharge port with respect to the orbiting angle $\theta$ are calculated based on the input parameters. Then the pressure $P_i$ and the temperature $T_i$ in the control volume at each angle $\theta_i$ are calculated for ideal working process and taken as initial values. Thirdly, the differential equations mentioned above can be solved simultaneously by 4 step Runge-Kutta method. After calculating the whole cycle, the deviation of pressure $P_i$ at each angle $\theta_i$ in this iteration and the value obtained from last iteration decided the convergence of the calculation.

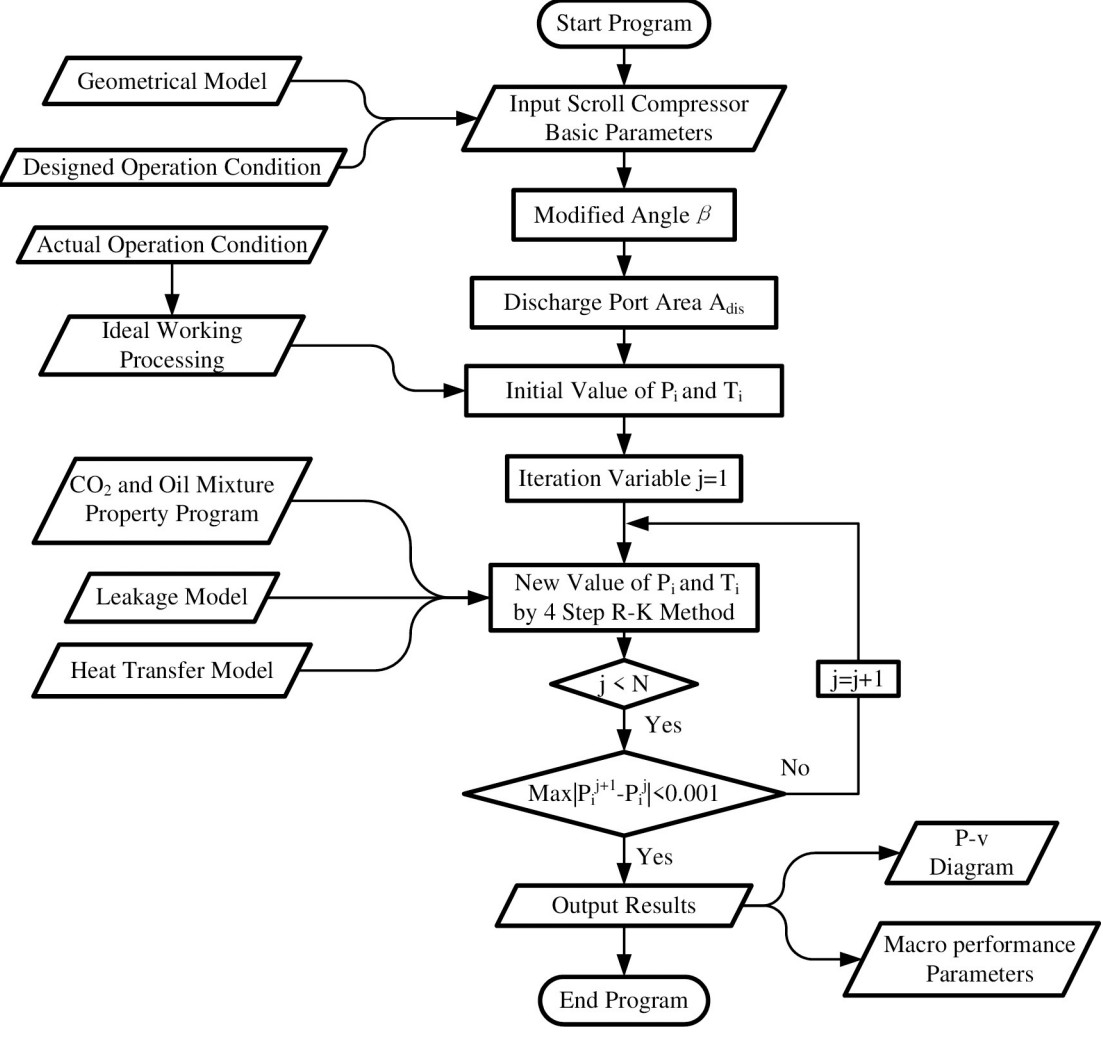

**Fig 5. Flow chart of simulation model.**

When the errors are not match with the desired value, the new data will be the initial values for the next iteration. Otherwise the iteration is convergent and the performance parameters and other results will be output.

## 4. Experimental study

A $CO_2$ refrigeration test rig is constructed in order to measure the performance of the prototype scroll compressor. As depicted in Fig 6, a measured compressor prototype, a gas cooler, an expansion valve and a superheater are arranged in an outdoor chamber. An evaporator is installed in another chamber named indoor chamber. Two air handled equipment units are used to control the dry bulb and wet bulb temperatures in the outdoor chamber and the indoor chamber, respectively. The condition of the compressor cabinet is also controlled.

Temperatures and pressures are measured by Pt100 thermometers (accuracy ±0.2°C) and pressure sensors (accuracy ±0.2% FS) at the important components inlet/outlet, such as compressor, gas cooler expansion valve and evaporator. The mass flow rate of refrigerant is obtained with a coriolis mass flow meter (accuracy ±0.2% FS) located downstream of the

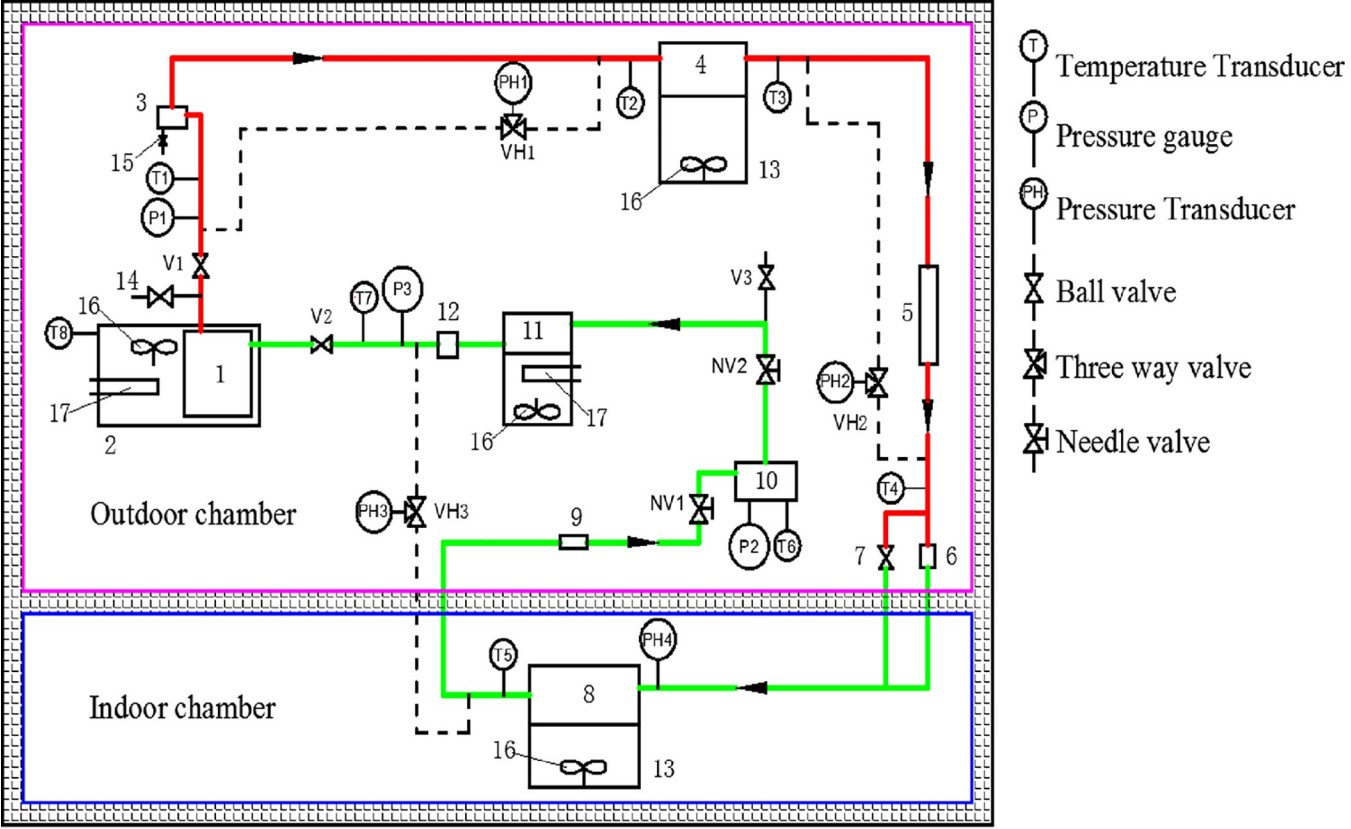

1.Compressor; 2.Compressor chamber; 3.Oil separator; 4.Gas cooler; 5.Filter; 6.Expansion valve; 7.Bypass valve; 8.Evaporator; 9.Glass sight; 10.Gas-liquid separator; 11.Superheater; 12.Mass flow meter; 13.Wind channel; 14.Safety valve; 15.Oil return valve; 16.Fan; 17.Electric heating

**Fig 6. Schematic diagram of experimental system.**

superheater. A three-phase electric power meter tests the compressor input power. Finally, all the measurements are real time acquired and displayed in a computer by a data acquisition system.

In operation, the pressure at the compressor outlet is kept at a desired value by the expansion valve, and the compressor inlet pressure is controlled by altering the speed of the evaporator fan and auxiliary adjusting the indoor temperature. The superheater is used to stably maintain the suction temperature of the compressor at a given temperature. When the temperatures and the pressures at the compressor inlet/outlet fluctuate less than ±0.5°C and ± 0.2% of reading simultaneously for 10 minutes, the compressor run stability can be achieved, and the data acquisition system begins to record test data. The final experimental data represent the average value of 60 minutes stable state acquisition.

## 5. Results and discussion

The prototype of oil-injection $CO_2$ hermetic scroll compressor is tested and simulated for various oil-injection quantities at the working conditions given in Table 2. The definition of oil-injection quantities is expressed by mass ratio of oil to gas. The simulation and experimental

**Table 2. Test conditions.**

| Items | Suction pressure / MPa | Suction superheat /°C | Discharge pressure /MPa | Compressor speed /(rev·min⁻¹) |
|---|---|---|---|---|
| Case 1 | 4.5,5.0 | 9 | 10.0 | 2860 |
| Case 2 | 3.9~5.0 | 9 | 10.0 | 2860 |
| Case 3 | 4.0 | 9 | 8.0~10.0 | 2860 |

results including mass flow rate of $CO_2$ refrigerant, power consumption, discharge temperature and efficiencies are compared and analyzed.

## 5.1 Effect of mass ratio of oil to refrigerant

Figs 7–11 exhibit the influences of mass ratio of oil to refrigerant on volumetric and indicated efficiencies, mass flow rate of $CO_2$, discharge temperature and input power under test 1 condition. It is shown that the simulation results agree well with the experimental data. The maximal deviations between experimental and simulate data are less than 10%, 15%, 10%, 9% and 2% in turn.

Observing the simulated curves of volumetric efficiency in Fig 7, the similar change trend as that of reference 16 is found. There is a peak volumetric efficiency exiting at a certain mass of ratio of oil to refrigerant called optimal oil-injection quantity. But the difference from reference 16 is that the curve in the range from zero to optimal oil-injection quantity is much steeper and the volumetric efficiency is much lower without oil-injection under similar operation conditions. This is because the prototype compressor has the larger leakage gap and the leakage loss is more sensitive to oil-injection quantity. Further analysis shows that the optimal oil-injection quantity decreases from 7.9% to 6.3% when the suction pressure increases from

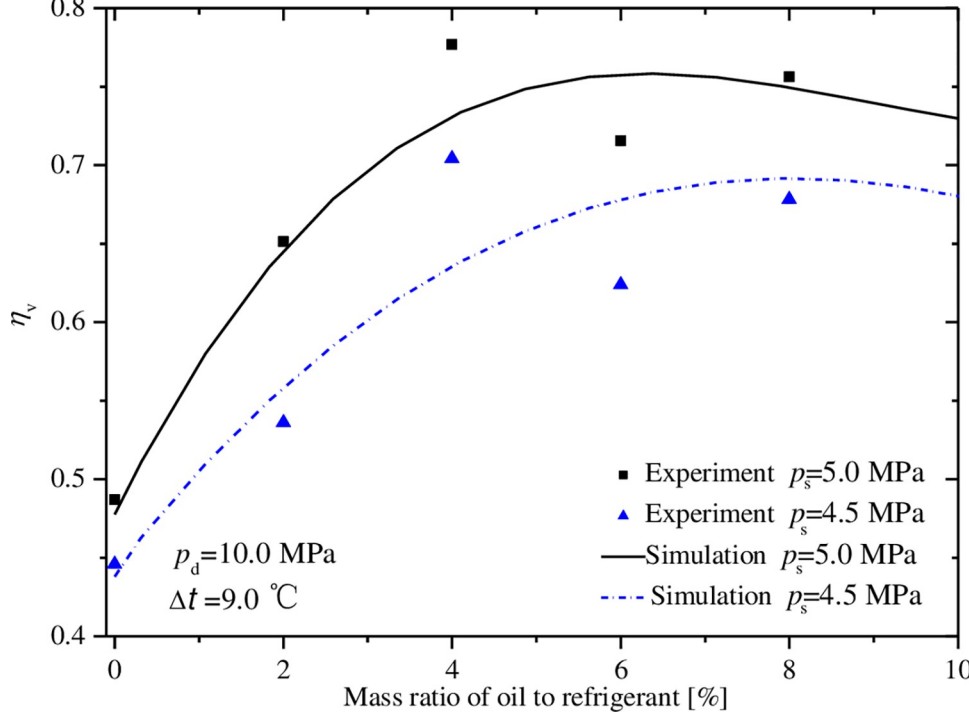

**Fig 7. Effect of mass ratio of oil to gas on volumetric efficiency.**

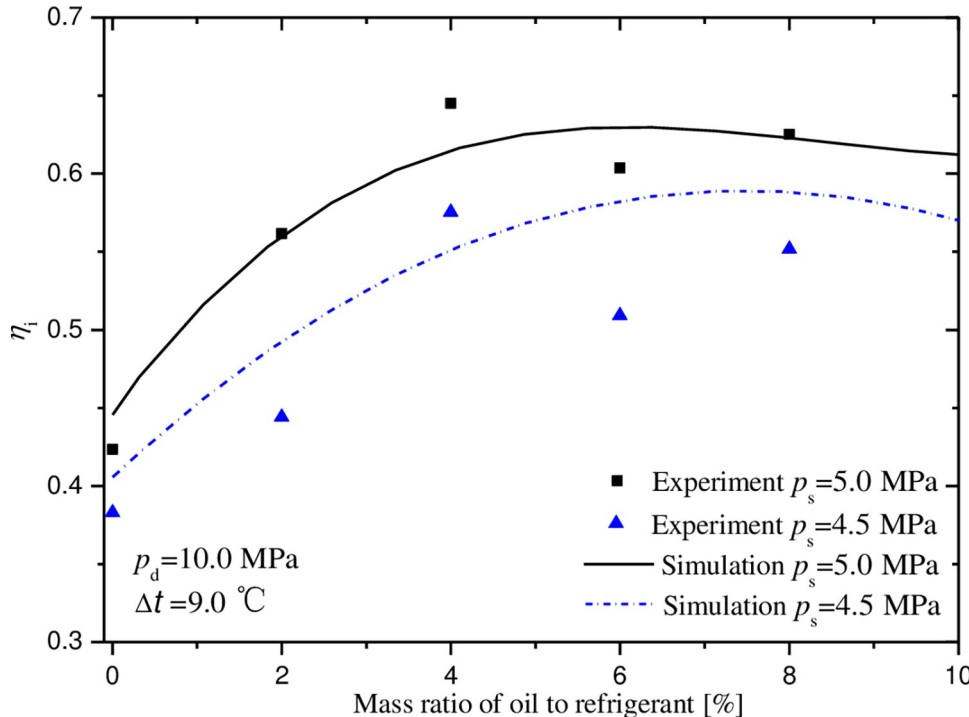

**Fig 8. Effect of mass ratio of oil to gas on indicated efficiency.**

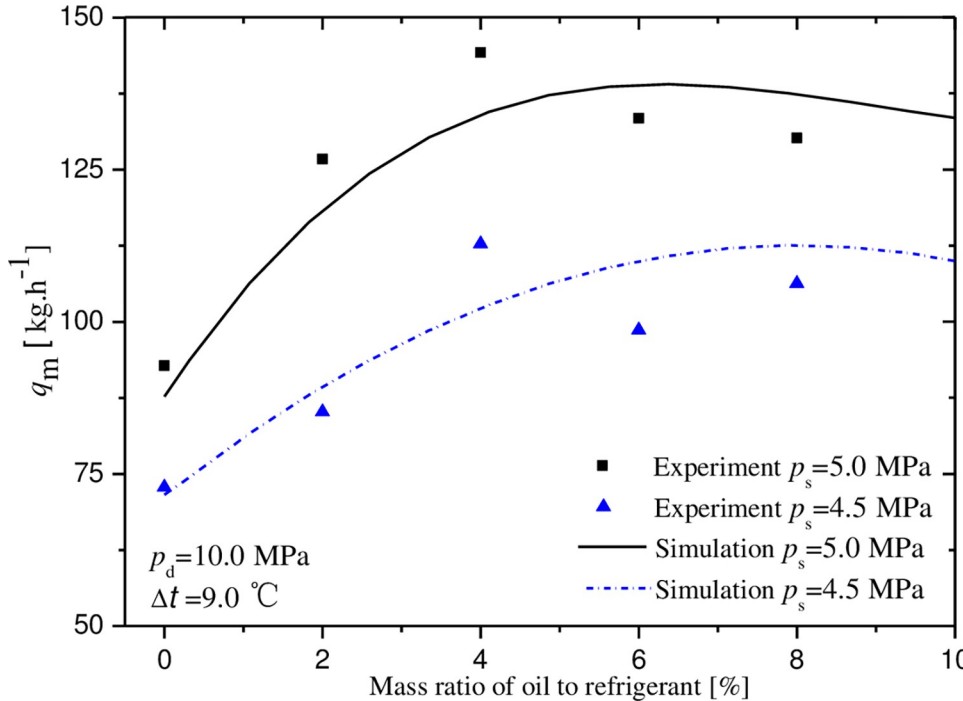

**Fig 9. Effect of mass ratio of oil to gas on mass flow rate.**

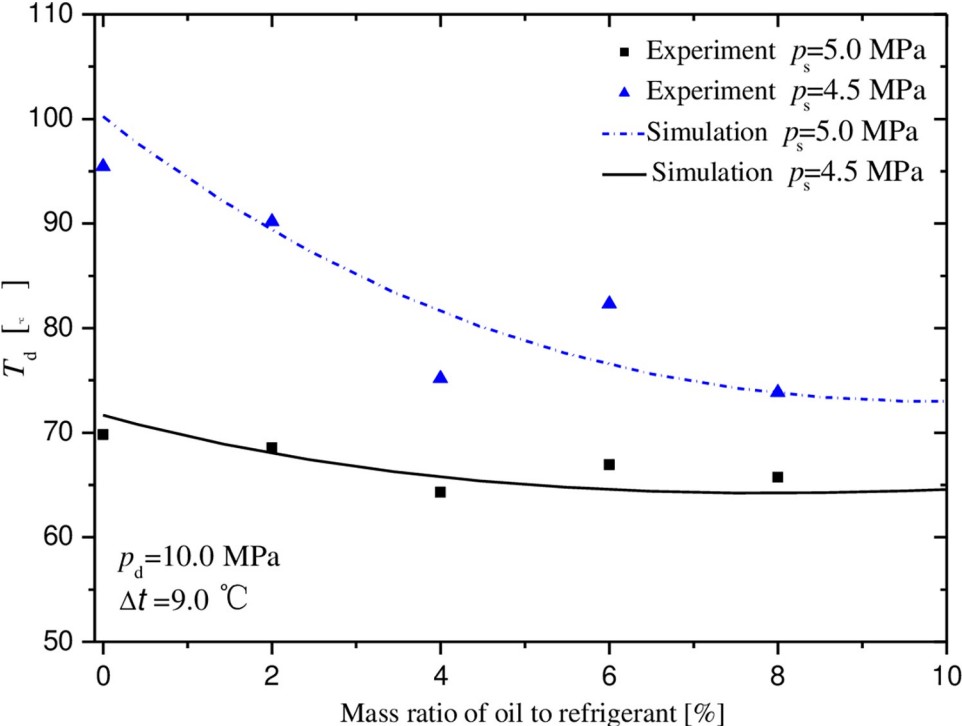

**Fig 10. Effect of mass ratio of oil to gas on discharge temperature.**

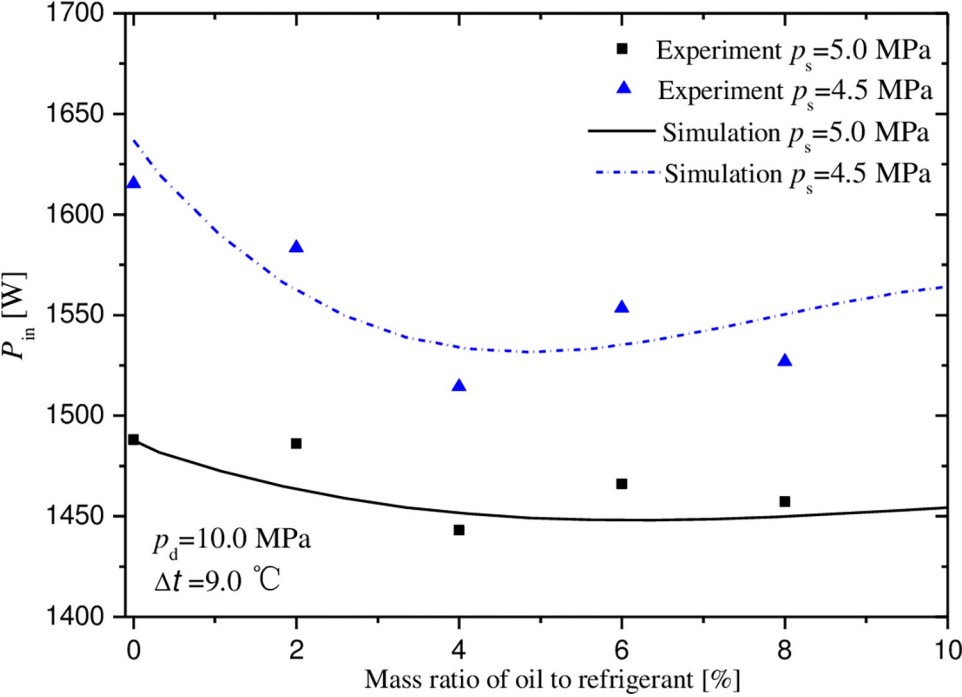

**Fig 11. Effect of mass ratio of oil to gas on power input.**

4.5Mpa to 5.0Mpa, and the volumetric efficiency grows a little faster under the suction pressure of 5.0MPa. The reason is that the velocity of refrigerant through leakage channel is lower under lower pressure difference, and a thicker oil film attaching on the wall of leakage channel formed more easily to prevent refrigerant from passing.

The variations of indicated efficiency and mass flow rate of $CO_2$ displayed in Figs 8 and 9 is similar to that of volumetric efficiency. At the points for optimal oil-injection quantity, the indicated efficiencies reach 58.9% and 63.0%, and the mass flow rates are 112.6 and 138.7 kg·h$^{-1}$, under the conditions of the suction pressures 4.5 and 5.0MPa.

Fig 10 shows the variations of discharge temperature with respect to mass ratio of oil to refrigerant. It is shown that the discharge temperature decreases with the increasing of mass ratio, and finally it tends to a steady temperature. The discharge temperature under the suction pressure of 4.5MPa is much higher than that under another condition for low mass ratio. For example, about the temperature difference of 25˚C is exhibited without oil-injection. Then the temperature difference reduces and drives to stability with increasing oil-injection quantity. Certainly, the faster falling down of discharge temperature can be found at the 4.5MPa suction pressure. These phenomena are caused by internal leakage between neighbor chambers. That is to say, the recompression of internal leakage gas will rise the discharge temperature. The higher pressure difference will cause the larger internal leakage, which leads to more serious recompression and raises further higher discharge temperature. When the approximately stable internal leakage is presented after optimal oil-injection quantity, the discharge temperature varies little.

Compared with the reference 16, it is found that the discharge temperature tends to a certain value rather than increases slowly, when mass ratio increasing continually after reaching optimal oil-injection quantity. This is due to different suction channel of the two compressor prototypes. The compressor in the reference 16 is high pressure shell and the refrigerant is sucked into the suction chamber directly, while the prototype in this paper is low pressure shell and the refrigerant flowing into the suction port firstly passes through the motor and is preheated before entering the suction chamber. Therefore, the effect of refrigerant heated by injected oil on discharge temperate is very small.

The input power variation as a function of mass ratio is exhibited in Fig 11. Different from the other parameter mentioned above, the input power decreases firstly and increase subsequently with the increase of mass ratio. Moreover, the minimums of input power are obtained at the mass ratio of 4.5% and 6.2% for the suction pressures of 4.5 and 5MPa, which are different from the optimal oil-injection quantities. It is a comprehensive result of the improvement in lubrication and leakage conditions. The prototype performance index in Figs 7–11 can be calculated by experimental data, which was shown by S1 Table.

## 5.2 Effect of pressure ratio

Figs 12–15 show the curves of volumetric efficiency, indicated efficiency, input power and refrigerant mass flow rate with respect to pressure ratio for fixed discharge pressure of 10.0MPa(case 2) and fixed suction pressure of 4.0MPa(case 3). Approximate linear relationship of the four parameters with pressure ratio can be obtained under the two cases. The volumetric and indicated efficiencies, the mass flow rate and the input power vary in range of 89.5% to 56.1%, 79.9% to 46.7%, 144.2˚C to 74.8˚C, 1167.4W to 1683.5W, respectively, when the pressure ratio changes from 2.0 to 2.6. However, some differences of the curves in the two cases can also be found by further analysis.

As shown in Figs 12 and 13, the volumetric and indicated efficiencies decrease faster with the increases of pressure ratio in case 3 than that in case 2. However, the refrigerant mass flow

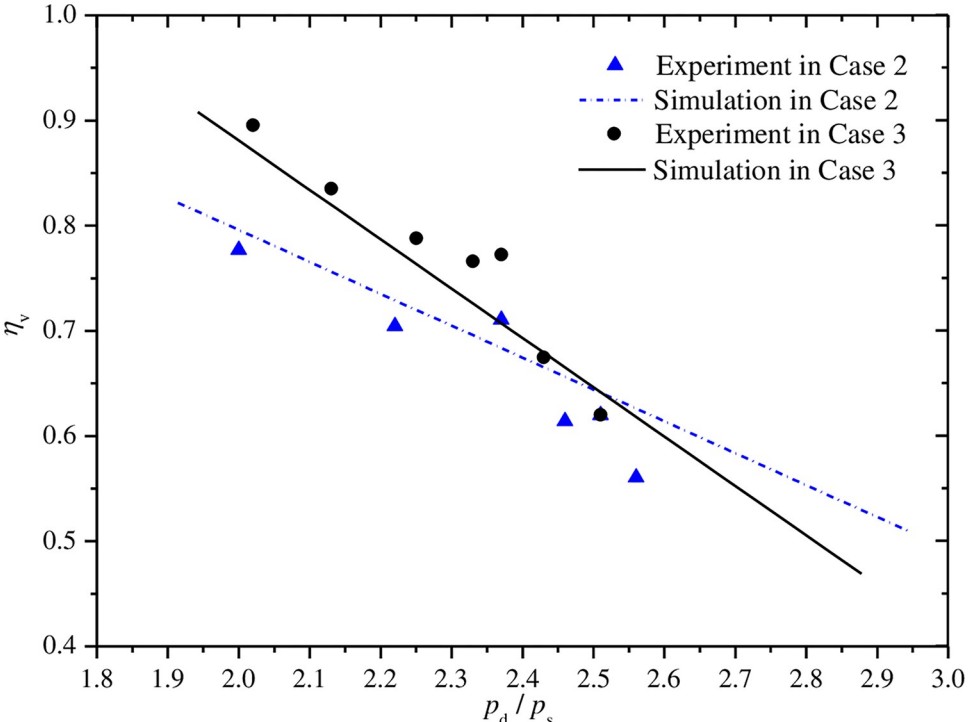

**Fig 12. Effect of pressure ratio on volumetric efficiency.**

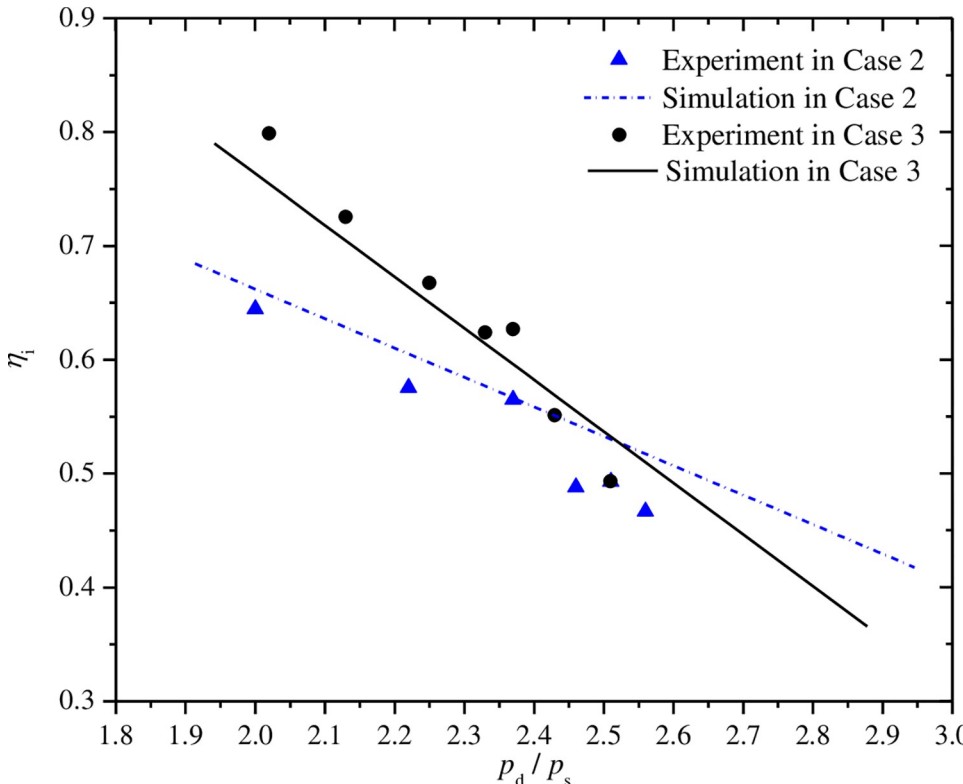

**Fig 13. Effect of pressure ratio on indicated efficiency.**

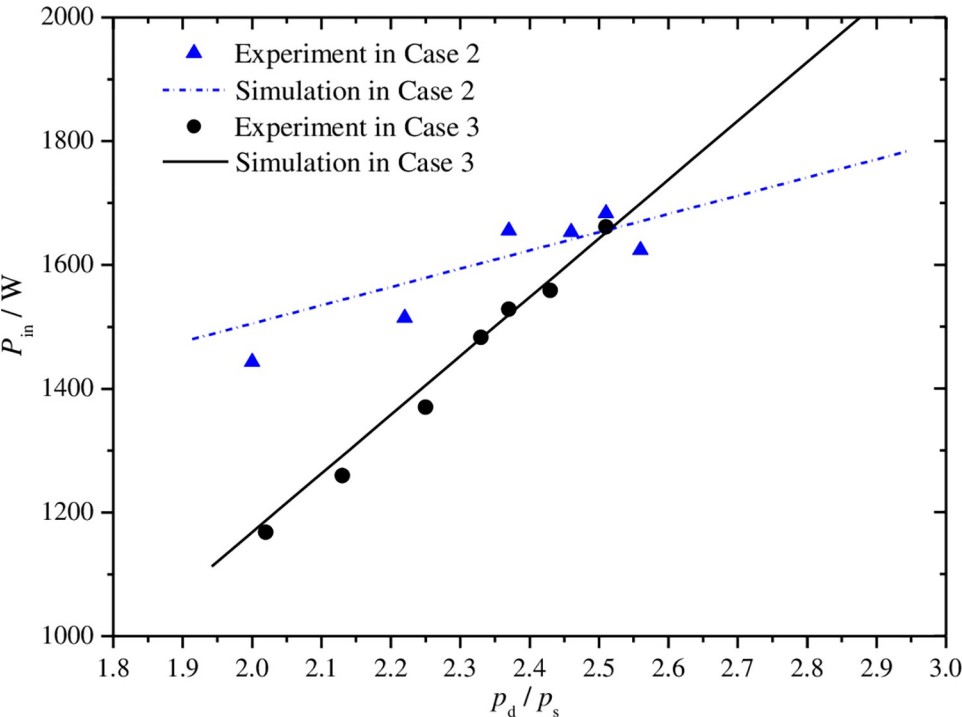

**Fig 14. Effect of pressure ratio on power input.**

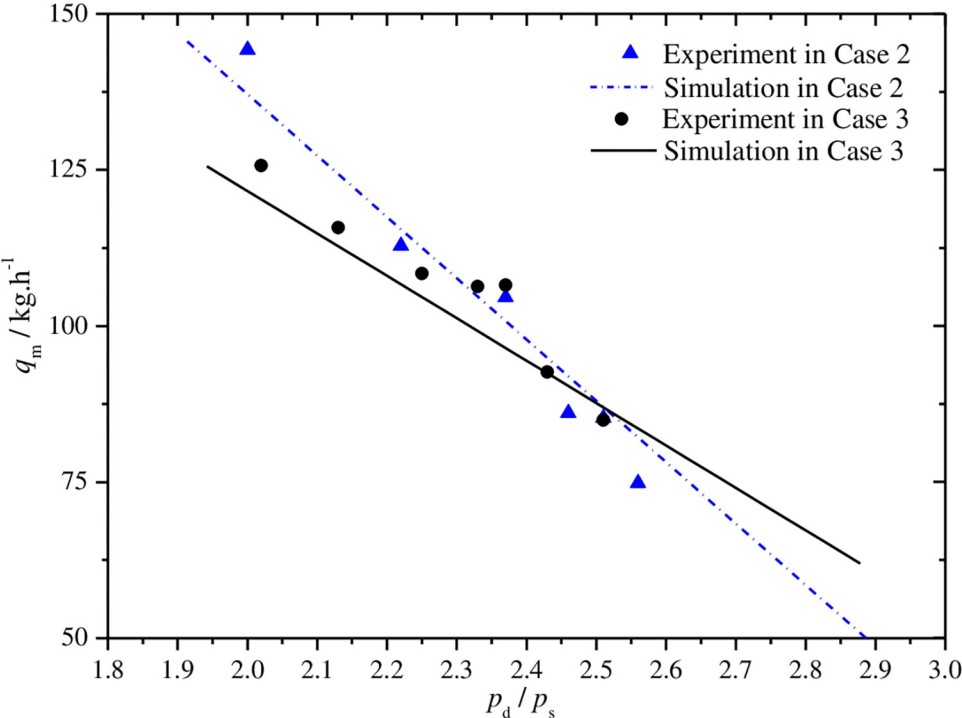

**Fig 15. Effect of pressure ratio on mass flow rate.**

rate reduces slightly faster in case 2 as shown in Fig 15. The analysis result represents that the faster increase of pressure difference between suction and discharge pressures with pressure ratio in case 3 is a major reason to explain the first phenomenon. The mass flow rate is influenced by special volume of refrigerant at compressor suction port beside leakage loss caused by the just mentioned pressure difference. In case 2, the refrigerant special volume at the suction port increases obviously, while the special volume is kept a constant value in another case. So the comprehensive result of the both above factors leads to the appearance of the second phenomenon. In Fig 14, the sharper slope of input power curve in case 3 is also the final result of the both factors.

## 5.3 Effect of mass ratio of oil to refrigerant on the pressure in working chamber

The pressure in a working chamber with respect to rotation angle is shown in Fig 16 and it is calculated for three mass ratios when the discharge/suction pressure and superheat temperature are 10.0/3.385MPa and 8.3°C, respectively. In order to analyze the influence of mass ratio on the pressure obviously and conveniently, the three curves are plotted in Fig 16. However, the solid line is pressure plotting for the operation without oil-injection called basic operation. The other two curves, represented by dash and dash dot lines, are pressure difference plotting for the operation injected mass ratio of 5% and 10% called oil-injection operation. The pressure difference is the result of the pressure in the basic operation subtracted from the pressure in the oil-injection operation at the same rotation angle.

From Fig 16, it is found that the effect of mass ratio on the pressure can be neglected during the steady suction and discharge processes. However, since the sealing between the fixed scroll and the orbiting scroll is well improved in the oil-injection operations, the visible change of the pressure is presented in the end of suction process and the compression process. From the

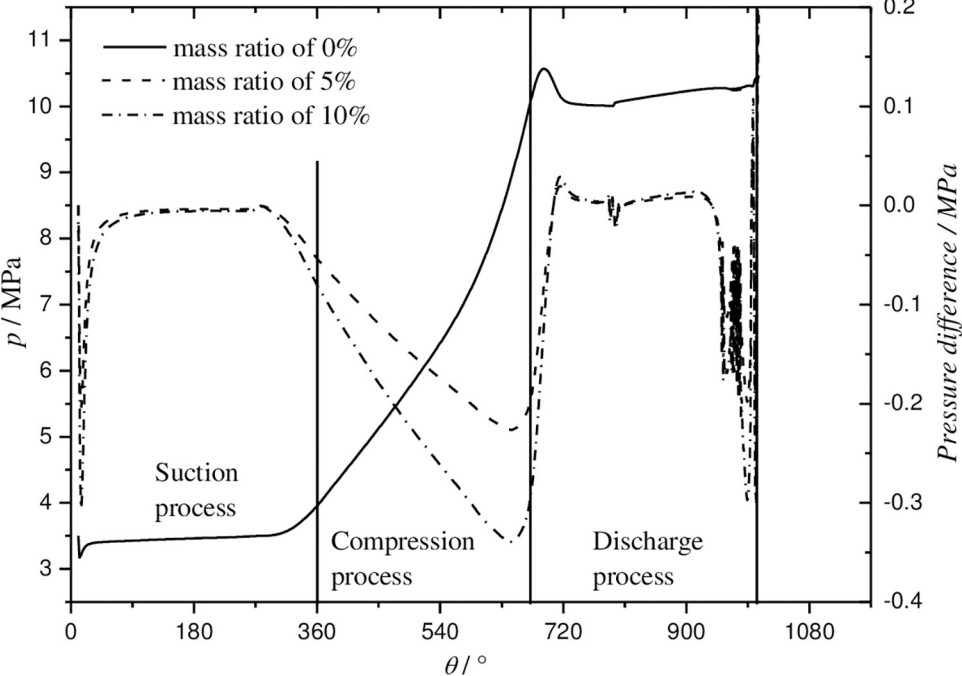

**Fig 16. Effect of mass ratio of oil to gas on pressure in working chamber.**

rotation angle of about 282˚ to 645˚, the pressure increases more slowly because of less leakage from higher pressure previous chamber, and then rises faster in the rest of the compression procession in the oil-injection operations due to less leakage to lower pressure next chamber. As a result, the pressure in this working chamber in the oil-injection operations is lower than that of the basic operation at the end of compression precession of about 672˚. Therefore, the salient of pressure in the beginning of discharge process exceeding the normal pressure of 10MPa will fall down. For the higher mass ration of oil-injection, the phenomena described above will present more obviously. Moreover, the fluctuation nearby the end of discharge process is easily caused by 4 step Runge-Kutta method itself when the control volume is very small.

In summary, it could be concluded that on the condition of the discharge pressure fixed at 10MPa, the density of suction gas and mass flow rate decreased as the suction pressure decreased, the volumetric efficiency, the indicated efficiency decreased as the pressure ratio increased, the discharge temperature and the input power increased as the pressure ratio increased. The increased oil-injection rate into working chamber could improve the volumetric efficiency due to the oil film sealing on the leakage passage, and it could decrease the discharge temperature due to the partial gas compression heat absorbed by the lubricant oil, but the disadvantage is higher input power to pump the cycle of lubricant oil.

## 6. Conclusion

A prototype of oil-injection hermetic $CO_2$ scroll compressor is developed and tested at various working conditions and different oil-injection quantity. A simulation model is also conducted considering internal leakage loss and heat transfer. Comparison the simulation and experimental results represents that the simulation result agree well with the experimental data. The maximal deviations between experimental and simulate data are less than 10%, 15%, 10%, 9% and 2% in volumetric efficiency, indicated efficiency, mass flow rate of $CO_2$, discharge temperature and input power. Based on the theoretical and experimental analyses, the following conclusion can be drawn.

Oil-injection is an effective measure to improve the volumetric and indicated efficiencies, mass flow rate of $CO_2$, reduce discharge temperature and input power. For a scroll compressor with larger clearance gap, leakage loss has more sensitive to oil-injection quantity. The optimal oil-injection quantities exist and vary at various working conditions. The optimal oil injection quantity decreases from 7.9% to 6.3% with the suction pressure increases from 4.5Mpa to 5.0Mpa, when the discharge pressure fixed at 10MPa and the superheat temperature of 9˚C. But the minimums of input power are obtained at the mass ratio of 4.5% and 6.2%, which are different from the optimal oil-injection quantities.

The relationships between pressure ratio and the above parameters of compressor is approximate linear. The volumetric and indicated efficiencies of the compressor are in the range of 89.5% to 56.1% and 79.9% to 46.7% respectively, when the pressure ratio increases from 2.0 to 2.6. Moreover, the lines exhibit different slopes in the two case of constant suction or discharge pressure.

Oil-injection has little influence on pressure of working chamber during suction and discharge process, but can slow up the rising of pressure in compression progress and reduce the salient of pressure at the end of compression process.

The future research direction will be focused on the optimal value between axial clearance and oil-injection quantity at different working conditions. The main research contents are as follows: The variation curve of axial clearance will be measured at fast cooling working condition by using dynamic displacement sensor. The visualization study of gas and liquid two-

phase flow state will be obtained by Doppler Laser Velocimeter at optimal oil-injection quantity. The p-V diagram will be measured so as to obtain the actual indicated power and determine the actual pressure on both sides of the leakage passage. The research objective is that the volumetric efficiency will be increased to 85% at designed working condition, and the overall volumetric efficiency could improve 10% compared to the current experimental results.

## Supporting information

**S1 Table. Performance index of prototype obtained in the research.**
(DOCX)

## Author Contributions

**Conceptualization:** Jingying Hao, Zhizhong Wang.

**Data curation:** Jingying Hao, Zhizhong Wang.

**Formal analysis:** Jingying Hao, Zhizhong Wang, Pengcheng Shu.

**Funding acquisition:** Jingying Hao.

**Investigation:** Jingying Hao, Pengcheng Shu.

**Methodology:** Jingying Hao.

**Project administration:** Jingying Hao.

**Resources:** Jingying Hao, Zhizhong Wang, Bo Zhang, Pengcheng Shu.

**Software:** Zhizhong Wang, Bo Zhang.

**Supervision:** Gaoxuan Bu.

**Validation:** Gaoxuan Bu, Bo Zhang, Pengcheng Shu.

**Visualization:** Gaoxuan Bu, Bo Zhang, Pengcheng Shu.

**Writing – original draft:** Jingying Hao.

**Writing – review & editing:** Jingying Hao, Pengcheng Shu.

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
