## [Decision Letter · Decision Letter 0]

20 Aug 2024

PONE-D-24-28168Performance Study of Oil-injection Hermetic CO2 Scroll Compressor for Automotive Air Conditioning SystemPLOS ONE

Dear Dr. Hao,

Thank you for submitting your manuscript to PLOS ONE. After careful consideration, we feel that it has merit but does not fully meet PLOS ONE’s publication criteria as it currently stands. Therefore, we invite you to submit a revised version of the manuscript that addresses the points raised during the review process.

We look forward to receiving your revised manuscript.

Kind regards,

Joy Nondy, Ph. D.

Academic Editor

PLOS ONE

Journal Requirements:

The work described in the paper was funded by the Program for Changjiang Scholars and Innovative

Research Team in University (IRT0746), Key Technology R&D Program (2009BAA17B00) and the

Program of Guangdong province (2007B090400079)

The work described in the paper was funded by the Program for Changjiang Scholars and Innovative

Research Team in University (IRT0746), Key Technology R&D Program (2009BAA17B00) and the

Program of Guangdong province (2007B090400079)

The work described in the paper was funded by the Program for Changjiang Scholars and Innovative

Research Team in University (IRT0746), Key Technology R&D Program (2009BAA17B00) and the

Program of Guangdong province (2007B090400079)

6. Please amend either the abstract on the online submission form (via Edit Submission) or the abstract in the manuscript so that they are identical.

Reviewers' comments:

Reviewer's Responses to Questions

**Comments to the Author**

1. Is the manuscript technically sound, and do the data support the conclusions?

Reviewer #1: Yes

Reviewer #2: Yes

2. Has the statistical analysis been performed appropriately and rigorously? 

Reviewer #1: Yes

Reviewer #2: Yes

3. Have the authors made all data underlying the findings in their manuscript fully available?

Reviewer #1: Yes

Reviewer #2: Yes

4. Is the manuscript presented in an intelligible fashion and written in standard English?

Reviewer #1: Yes

Reviewer #2: Yes

5. Review Comments to the Author

**Reviewer #1:** The study investigated an oil injected CO2 scroll compressor. The performance metrices of the compressor were evaluated with various oil injection ratio. The research gives some interesting results and makes a valid contribution to the field of compressor engineering. However, some improvement can be made.

1. Nomenclature: Heat flow rate into the control volume: Q with dot is supposed, while "&" is given.

2. The introduction is supposed to include the newest research in oil injection technology for scroll compressor.

1. Wang C, Zhao Z, Wu J. Nearly isothermal compression characteristics of the helium oil-flooded scroll compressor. Applied Thermal Engineering. 2024;251:123597. doi:10/gtx989

2. James NA, Braun JE, Groll EA, Horton WT. Semi-empirical modeling and analysis of oil flooded R410A scroll compressors with liquid injection for use in vapor compression systems. International Journal of Refrigeration. 2016;66:50-63. doi:10/f8t7dh

3. How the gaps are determined in Table 1 should be clarified. Measured or Designed, or Assumed?

4. Sec. 3.4 The value of empirical formula for friction coefficient is supposed to be given.

5. Research perspective of oil injection method is suggested to discussed in Sec. 6

**Reviewer #2:** I have read the manuscript titled “Performance Study of Oil-injection Hermetic CO2 Scroll Compressor for Automotive Air Conditioning System” and found it to be a valuable contribution to the field. This research addresses an important topic, and the results presented are insightful. However, I believe there are certain areas where the manuscript could be further improved to enhance its clarity, depth, and overall impact.

1. Please avoid using abbreviations (e.g., A/C) in the abstract. Abbreviations can reduce clarity, especially for readers who may not be familiar with them.

2. The novelty and main contributions of this study in comparison to previous research are not clearly articulated. Please provide a more detailed description of these aspects in the Introduction section to better highlight the unique value and significance of this work.

3. The manuscript currently cites only three papers from the 2020s, with the majority of references being quite dated. Please consider adding more recent papers on the same topic to ensure the references are up-to-date and reflective of the current state of research.

4. The authors are advised to include a paragraph at the end of the conclusion that outlines potential future research directions on this topic. This addition would provide a more comprehensive sense of the article's significance to the readers.

5. In Sections 5.1-5.3, the authors have presented the variations quantitatively. It is recommended that the authors also provide a qualitative explanation of the parametric results. They should discuss the underlying physics behind the observed variations in the parameters. Why are these changes occurring?

6. The authors are advised to cite all equations used in the manuscript to provide appropriate references and enhance the credibility of the work.

7. Please increase the size of Figures 3 and 6 to ensure that all details are clearly visible and easily readable.

8. In Figures 8-16, there is inconsistency in the formatting of the labels. Some figures use bold labels, while others do not. To maintain uniformity throughout the document, I recommend standardizing the labelling style across all figures, either by making all labels bold or unbold, depending on the preferred style.

6. PLOS authors have the option to publish the peer review history of their article (what does this mean?). If published, this will include your full peer review and any attached files.

Reviewer #1: No

Reviewer #2: No

---

## [Author Response · Author response to Decision Letter 0]

12 Sep 2024

Reviewer #1: The study investigated an oil injected CO2 scroll compressor. The performance metrices of the compressor were evaluated with various oil injection ratio. The research gives some interesting results and makes a valid contribution to the field of compressor engineering. However, some improvement can be made.

1. Nomenclature: Heat flow rate into the control volume: Q with dot is supposed, while "&" is given.

Answer：Q with dot was changed to be Q without dot so as to easy to understand its mean.

2. The introduction is supposed to include the newest research in oil injection technology for scroll compressor.

Answer：Four papers of oil injection technology have been added so as to learn the latest development.

[19] Bell IH , Groll EA , Braun JE ,et al. Experimental Testing of an Oil-Flooded Hermetic Scroll Compressor[J]. International Journal of Refrigeration, 2013, 36(7):1866-1873. 

[20] Ramaraj Sugirdhalakshmi, Yang Bin, Braun James, et al. Experimental analysis of oil flooded R410A scroll compressor[J]. International Journal of Refrigeration, 2014; 46:185-195.

[21] James NA, Braun JE, Groll EA, et al. Semi-empirical modeling and analysis of oil flooded R410A scroll compressors with liquid injection for use in vapor compression systems[J]. International Journal of Refrigeration. 2016;66:50-63. doi:10/f8t7dh

[22] Wang C, Zhao Z, Wu J. Nearly isothermal compression characteristics of the helium oil-flooded scroll compressor[J]. Applied Thermal Engineering. 2024;251:123597. doi:10/gtx989.

3. How the gaps are determined in Table 1 should be clarified. Measured or Designed, or Assumed?

Answer：The designed value of axial and radial gap was 15 μm by considering the assembly precision and the effect of oil-injection to working chamber.

4. Sec. 3.4 The value of empirical formula for friction coefficient is supposed to be given.

Answer：The modified contents are presented as follows:

In the empirical formula, the force F was the force value of main bearing, sub bearing and crank pin respectively, which was obtained by the dynamic Characteristic analysis of Scroll Compressor[33], the value of friction coefficient was given by the volume 2 of mechanical design manual.

 (20)

5. Research perspective of oil injection method is suggested to discussed in Sec. 6

Answer：The modified contents are presented as follows:

In summary, it could be concluded that on the condition of the discharge pressure fixed at 10MPa, the density of suction gas and mass flow rate decreased as the suction pressure decreased, the volumetric efficiency, the indicated efficiency decreased as the pressure ratio increased, the discharge temperature and the input power increased as the pressure ratio increased. The increased oil-injection rate into working chamber could improve the volumetric efficiency due to the oil film sealing on the leakage passage, and it could decrease the discharge temperature due to the partial gas compression heat absorbed by the lubricant oil, but the disadvantage is higher input power to pump the cycle of lubricant oil.

Reviewer #2: I have read the manuscript titled “Performance Study of Oil-injection Hermetic CO2 Scroll Compressor for Automotive Air Conditioning System” and found it to be a valuable contribution to the field. This research addresses an important topic, and the results presented are insightful. However, I believe there are certain areas where the manuscript could be further improved to enhance its clarity, depth, and overall impact.

1. Please avoid using abbreviations(e.g., A/C) in the abstract. Abbreviations can reduce clarity, especially for readers who may not be familiar with them.

Answer：A/C abbreviations are modified as Air Conditioning in the whole paper.

2. The novelty and main contributions of this study in comparison to previous research are not clearly articulated. Please provide a more detailed description of these aspects in the Introduction section to better highlight the unique value and significance of this work.

Answer：The highlight of this work was updated.

In this paper, in order to solve the working process internal leakage of CO2 scroll compressor at the bigger differential pressure about 5 and 6 MPa, an oil supply control system is designed, which has effective distribution function in lubricating friction pairs and oil-injection to working chamber. A prototype of oil-injection CO2 scroll compressor is fabricated for automotive Air Conditioning. Experimental investigation on the prototype has been carried out at different conditions in order to test the effect of oil-injection on the prototype performance, so the change of volumetric efficiency, indicated efficiency and discharge temperature with respect to mass ratio of oil to refrigerant were conducted，the values of optimal oil-injection quantity at different working conditions were obtained. Moreover, a numerical simulation model involving gas leakage, and heat transfer is established and verified experimentally. Simulation results of prototype working process with different mass ratio of oil to refrigerant were presented.

3. The manuscript currently cites only three papers from the 2020s, with the majority of references being quite dated. Please consider adding more recent papers on the same topic to ensure the references are up-to-date and reflective of the current state of research.

Answer：Four papers of oil injection technology have been added so as to learn the latest development. Four papers about thermodynamic and dynamic characteristic model were added.

[19] Bell IH , Groll EA , Braun JE ,et al. Experimental Testing of an Oil-Flooded Hermetic Scroll Compressor[J]. International Journal of Refrigeration, 2013, 36(7):1866-1873. 

[20] Ramaraj Sugirdhalakshmi, Yang Bin, Braun James, et al. Experimental analysis of oil flooded R410A scroll compressor[J]. International Journal of Refrigeration, 2014; 46:185-195.

[21] James NA, Braun JE, Groll EA, et al. Semi-empirical modeling and analysis of oil flooded R410A scroll compressors with liquid injection for use in vapor compression systems[J]. International Journal of Refrigeration. 2016;66:50-63. doi:10/f8t7dh

[22] Wang C, Zhao Z, Wu J. Nearly isothermal compression characteristics of the helium oil-flooded scroll compressor[J]. Applied Thermal Engineering. 2024;251:123597. doi:10/gtx989.

[30] Chen Y, Halm NP Braun JE, et al. Mathematical modeling of scroll compressors - part II: overall scroll compressor modeling[J]. International Journal of Refrigeration, 2002, 25 (6): 751-764.

[31] Wang J, Martin B, Liu M, et al. A comprehensive study on a novel transcritical CO2 heat pump for simultaneous space heating and cooling – Concepts and initial performance[J]. Energy Conversion and Management, 2021. 243:114397.

[32] Hao JY, Wang ZZ, Zhao YY, et al. Research on the Dynamic Characteristic of CO2 Scroll Compressor[J]. Journal of Refrigeration, 2011,32(05):42-46.

[33] Kim K, Hong G ,Jang GH .Dynamic analysis of a flexible shaft in a scroll compressor considering solid contact and oil film pressure in journal bearings[J].International Journal of Refrigeration, 2021, 127(1):165-173.

4. The authors are advised to include a paragraph at the end of the conclusion that outlines potential future research directions on this topic. This addition would provide a more comprehensive sense of the article's significance to the readers.

Answer：The modified contents are presented as follows:

The future research direction will be focused on the optimal value between axial clearance and oil-injection quantity at different working conditions. The main research contents are as follows: The variation curve of axial clearance will be measured at fast cooling working condition by using dynamic displacement sensor. The visualization study of gas and liquid two-phase flow state will be obtained by Doppler Laser Velocimeter at optimal oil-injection quantity. The p-V diagram will be measured so as to obtain the actual indicated power and determine the actual pressure on both sides of the leakage passage. The research objective is that the volumetric efficiency will be increased to 85% at designed working condition, and the overall volumetric efficiency could improve 10% compared to the current experimental results.

5. In Sections 5.1-5.3, the authors have presented the variations quantitatively. It is recommended that the authors also provide a qualitative explanation of the parametric results. They should discuss the underlying physics behind the observed variations in the parameters. Why are these changes occurring?

Answer：The modified contents are listed as following:

In summary, it could be concluded that on the condition of the discharge pressure fixed at 10MPa, the density of suction gas and mass flow rate decreased as the suction pressure decreased, the volumetric efficiency, the indicated efficiency decreased as the pressure ratio increased, the discharge temperature and the input power increased as the pressure ratio increased. The increased oil-injection rate into working chamber could improve the volumetric efficiency due to the oil film sealing on the leakage passage, and it could decrease the discharge temperature due to the partial gas compression heat absorbed by the lubricant oil, but the disadvantage is higher input power to pump the cycle of lubricant oil.

6. The authors are advised to cite all equations used in the manuscript to provide appropriate references and enhance the credibility of the work.

Answer：Relative research papers for all equations were added in this paper.

7. Please increase the size of Figures 3 and 6 to ensure that all details are clearly visible and easily readable.

Answer：The size of figures 5 is enlarged. The line type of device and pipeline are changed to bold so as to clearly display the flowchart of the CO2 refrigeration test rig in figures 6.

8. In Figures 8-16, there is inconsistency in the formatting of the labels. Some figures use bold labels, while others do not. To maintain uniformity throughout the document, I recommend standardizing the labelling style across all figures, either by making all labels bold or unbold, depending on the preferred style.

Answer：the size and unbold label of figures 8-16 are modified using the same format.

---

## [Editor Report · Decision Letter 1]

16 Sep 2024

Performance Study of Oil-injection Hermetic CO2 Scroll Compressor for Automotive Air Conditioning System

PONE-D-24-28168R1

Dear Dr. Hao,

We’re pleased to inform you that your manuscript has been judged scientifically suitable for publication and will be formally accepted for publication once it meets all outstanding technical requirements.

Kind regards,

Joy Nondy, Ph. D.

Academic Editor

PLOS ONE
---

## [Editor Report · Acceptance letter]

30 Sep 2024

PONE-D-24-28168R1 

PLOS ONE

Dear Dr. Hao, 

I'm pleased to inform you that your manuscript has been deemed suitable for publication in PLOS ONE. Congratulations! Your manuscript is now being handed over to our production team.

Kind regards, 

on behalf of

Dr. Joy Nondy 

Academic Editor

PLOS ONE